# OvA-INN: Continual Learning with Invertible Neural Networks

## Abstract

In the field of Continual Learning, the objective is to learn several tasks one after the other without access to the data from previous tasks. Several solutions have been proposed to tackle this problem but they usually assume that the user knows which of the tasks to perform at test time on a particular sample, or rely on small samples from previous data and most of them suffer of a substantial drop in accuracy when updated with batches of only one class at a time. In this article, we propose a new method, OvA-INN, which is able to learn one class at a time and without storing any of the previous data. To achieve this, for each class, we train a specific Invertible Neural Network to extract the relevant features to compute the likelihood on this class. At test time, we can predict the class of a sample by identifying the network which predicted the highest likelihood. With this method, we show that we can take advantage of pretrained models by stacking an Invertible Network on top of a features extractor. This way, we are able to outperform state-of-the-art approaches that rely on features learning for the Continual Learning of MNIST and CIFAR-100 datasets. In our experiments, we reach 72% accuracy on CIFAR-100 after training our model one class at a time.

## 1 Introduction

A typical Deep Learning workflow consists in gathering data, training a model on this data and finally deploying the model in the real world (Goodfellow et al., 2016). If one would need to update the model with new data, it would require to merge the old and new data and process a training from scratch on this new dataset. Nevertheless, there are circumstances where this method may not apply. For example, it may not be possible to store the old data because of privacy issues (health records, sensible data) or memory limitations (embedded systems, very large datasets). In order to address those limitations, recent works propose a variety of approaches in a setting called **Continual Learning** (Parisi et al., 2018).

In Continual Learning, we aim to learn the parameters $w$ of a model on a sequence of datasets $\mathcal{D}_i = \{(x_i^j, y_i^j)\}_{j=1}^{n_i}$ with the inputs $x_i^j \in \mathcal{X}^i$ and the labels $y_i^j \in \mathcal{Y}^i$, to predict $p(y^*|w, x^*)$ for an unseen pair $(x^*, y^*)$. The training has to be done on each dataset, one after the other, without the possibility to reuse previous datasets. The performance of a Continual Learning algorithm can then be measured with two protocols : *multi-head* or *single-head*. In the multi-head scenario, the task identifier $i$ is known at test time. For evaluating performances on task $i$, the set of all possible labels is then $\mathcal{Y} = \mathcal{Y}^i$. Whilst in the single-head scenario, the task identifier is unknown, in that case we have $\mathcal{Y} = \cup_{i=1}^N \mathcal{Y}^i$ with $N$ the number of tasks learned so far. For example, let us say that the goal is to learn MNIST sequentially with two batches: using only the data from the first five classes and then only the data from the remaining five other classes. In multi-head learning, one asks at test time to be able to recognize samples of 0-4 among the classes 0-4 and samples of 5-9 among classes 5-9. On the other hand, in single-head learning, one can not assume from which batch a sample is coming from, hence the need to be able to recognize any samples of 0-9 among classes 0-9. Although the former one has received the most attention from researchers, the last one fits better to the desiderata of a Continual Learning system as expressed in Farquhar & Gal (2018) and (van de Ven & Tolias, 2019). The single-head scenario is also notoriously harder than its multi-head counterpart (Chaudhry et al., 2018) and is the focus of the present work.

Updating the parameters with data from a new dataset exposes the model to drastically deteriorate its performance on previous data, a phenomenon known as *catastrophic forgetting* (McCloskey & Cohen, 1989). To alleviate this problem, researchers have proposed a variety of approaches such as storing a few samples from previous datasets (Rebuffi et al., 2017), adding *distillation* regularization (Li & Hoiem, 2018), updating the parameters according to their usefulness on previous datasets (Kirkpatrick et al., 2017), using a generative model to produce samples from previous datasets (Kemker & Kanan, 2017). Despite those efforts toward a more realistic setting of Continual Learning, one can notice that, most of the time, results are proposed in the case of a sequence of batches of multiple classes. This scenario often ends up with better accuracy (because the learning procedure highly benefits of the diversity of classes to find the best tuning of parameters) but it does not illustrate the behavior of those methods in the worst case scenario. In fact, Continual Learning algorithms should be robust in the size of the batch of classes.

In this work, we propose to implement a method specially designed to handle the case where each task consists of only one class. It will therefore be evaluated in the single-head scenario. Our approach, named One-versus-All Invertible Neural Networks (OvA-INN), is based on an invertible neural network architecture proposed by Dinh et al. (2014). We use it in a One-versus-All strategy : each network is trained to make a prediction of a class and the most confident one on a sample is used to identify the class of the sample. In contrast to most other methods, the training phase of each class can be independently executed from one another.

The contributions of our work are : *(i)* a new approach for Continual Learning with one class per batch; *(ii)* a neural architecture based on Invertible Networks that does not require to store any of the previous data; *(iii)* state-of-the-art results on several tasks of Continual Learning for Computer Vision (CIFAR-100, MNIST) in this setting.

We start by reviewing the closest methods to our approach in Section 2, then explain our method in Section 3, analyse its performances in Section 4 and identify limitations and possible extensions in Section 5.

## 2 RELATED WORK

**Generative models** Inspired by biological mechanisms such as the hippocampal system that rapidly encodes recent experiences and the memory of the neocortex that is consolidated during sleep phases, a natural approach is to produce samples of previous data that can be added to the new data to learn a new task. FearNet (Kemker & Kanan, 2017) relies on an architecture based on an autoencoder, whereas Deep Generative Replay (Shin et al., 2017) and Parameter Generation and Model Adaptation (Hu et al., 2018) propose to use a generative adversarial network. Those methods present good results but require complex models to be able to generate reliable data. Furthermore, it is difficult to assess the relevance of the generated data to conduct subsequent training iterations.

**Coreset-based models** These approaches alleviate the constraint on the availability of data by allowing the storage of a few samples from previous data (which are called *coreset*). iCaRL (Rebuffi et al., 2017) and End-to-end IL (Castro et al., 2018) store 2000 samples from previous batches and rely on respectively a distillation loss and a mixture of cross-entropy and distillation loss to alleviate forgetting. The authors of SupportNet (Li et al., 2018) have also proposed a strategy to select relevant samples for the coreset. Gradient Episodic Memory (Lopez-Paz et al., 2017) ensures that gradients computed on new tasks do not interfere with the loss of previous tasks. Those approaches give the best results for single-head learning. But, similarly to generated data, it is not clear which data may be useful to conduct further training iterations. In this paper, we are challenging the need of the coreset for single-head learning.

**Distance-based models** These methods propose to embed the data in a space which can be used to identify the class of a sample by computing a distance between the embedding of the sample and a reference for each class. Among the most popular, we can cite Matching Networks (Vinyals et al., 2016) and Prototypical Networks (Snell et al., 2017), but these methods have been mostly applied to few-shot learning scenarios rather than continual.

**Regularization-based approaches**  These approaches present an attempt to mitigate the effect of catastrophic forgetting by imposing some constraints on the loss function when training subsequent classes. Elastic Weight Consolidation (Kirkpatrick et al., 2017), Synaptic Intelligence (Zenke et al., 2017) and Memory Aware Synapses (Aljundi et al., 2018) all seek to prevent the update of weights that were the most useful to discriminate between previous classes. Hence, it is possible to constrain the learning of a new task in such a way that the most relevant weights for the previous tasks are less susceptible to be updated. Learning without forgetting (Li & Hoiem, 2018) proposes to use knowledge distillation to preserve previous performances. The network is divided in two parts : the shared weights and the dedicated weights for each task. When learning a new task A, the data of A get assigned "soft" labels by computing the output by the network with the dedicated weight for each previous task. Then the network is trained with the loss of task A and is also constrained to reproduce the recorded output for each other tasks. In Rannen et al. (2017), the authors propose to use an autoencoder to reconstruct the extracted features for each task. When learning a new task, the features extractor is adapted but has to make sure that the autoencoder of the other tasks are still able to reconstruct the extracted features from the current samples. While these methods obtain good results for learning one new task, they become limited when it comes to learn several new tasks, especially in the one class per batch setting.

**Expandable models**  In the case of the multi-head setting, it has been proposed to use the previously learned layers and complete them with new layers trained on a new task. This strategy is presented in Progressive Networks (Rusu et al., 2016). In order to reduce the growth in memory caused by the new layers, the authors of Dynamically Expandable Networks (Yoon et al., 2018) proposed an hybrid method which retrains some of the previous weights and add new ones when necessary. Although these approaches work very well in the case of multi-head learning, they can not be adapted to single-head and are therefore not included in benchmarks with OvA-INN.

## 3    CLASS-BY-CLASS CONTINUAL LEARNING WITH INVERTIBLE NETWORKS

### 3.1    MOTIVATIONS AND CHALLENGE

We investigate the problem of training several datasets in a sequential fashion with batches of only one class at a time. Most approaches of the state-of-the-art rely on updating a features extractor when data from a new class is available. But this strategy is unreliable in the special case we are interested in, namely batches of data from only one class. With few or no sample of negative data, it is very inefficient to update the weights of a network because the setting of deep learning normally involves vast amounts of data to be able to learn to extract valuable features. Without enough negative samples, the training is prone to overfit the new class. Recent works have proposed to rely on generative models to overcome this lack of data by generating samples of old classes. Nevertheless, updating a network with sampled data is not as efficient as with real data and, on the long run, the generative quality of early classes suffer from the multiple updates.

### 3.2    OUT-OF-DISTRIBUTION DETECTION FOR CONTINUAL LEARNING

Our approach consists in interpreting a Continual Learning problem as several out-of-distribution (OOD) detection problems. OOD detection has already been studied for neural networks and can be formulated as a binary classification problem which consists in predicting if an input $x$ was sampled from the same distribution as the training data or from a different distribution (Lee et al., 2017; Liang et al., 2017). Hence, for each class, we can train a network to predict if an input $x$ is likely to have been sampled from the distribution of this class. The class with the highest confidence can be used as the prediction of the class of $x$. This training procedure is particularly suitable for Continual Learning since the training of each network does not require any negative sample.

Using the same protocol as NICE (Dinh et al., 2014), for a class $i$, it is possible to train a neural network $f_i$ to fit a prior distribution $p$ and compute the exact log-likelihood $l_i$ on a sample $x$ :

$$l_i(x) = \log(p(f_i(x)) \tag{1}$$

To obtain the formulation of log-likelihood as expressed in Equation 1, the network $f_i$ has to respect some constraints discussed in Section 3.3. Keeping the same hypothesis as NICE, we consider the

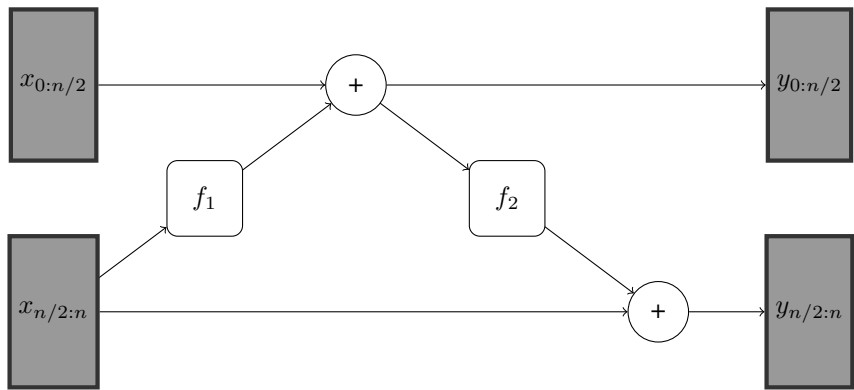

Figure 1: Forward pass in an invertible block. $x$ is split in $x_{0:n/2}$ and $x_{n/2:n}$. $f_1$ and $f_2$ can be any type of Neural Networks as long as the dimension of their output dimension is the same as their input dimension. In our experiments, we stack two of these blocks one after the other and use fully-connected feedforward layers for $f_1$ and $f_2$.

case where $p$ is a distribution with independent components $p_d$ :

$$p(f_i(x)) = \prod_d p_d(f_{i,d}(x)) \tag{2}$$

In our experiments, we considered $p_d$ to be standard normal distributions. Although, it is possible to learn the parameters of the distributions, we found experimentally that doing so decreases the results. Under these design choices, the computation of the log-likelihood becomes :

$$l_i(x) = \sum_d \log(p_d(f_{i,d}(x)) = -\sum_d \frac{1}{2} f_{i,d}(x)^2 + \sum_d \log\left(\frac{1}{\sqrt{2\pi}}\right) = -\frac{1}{2}\|f_i(x)\|_2^2 + \beta \tag{3}$$

where $\beta = -n \log\left(\sqrt{2\pi}\right)$ is a constant term.

Hence, identifying the network with the highest log-likelihood is equivalent to find the network with the smallest output norm.

### 3.3   INVERTIBLE NEURAL NETWORKS

The neural network architecture proposed by NICE is designed to operate a change of variables between two density functions. This assumes that the network is invertible and respect some constraints to make it efficiently computable.

An invertible block (see Figure 1) consists in splitting the input $x$ into two subvectors $x_1$ and $x_2$ of equal size; then successively applying two (non necessarily invertible) networks $f_1$ and $f_2$ following the equation :

$$\begin{cases} y_1 = f_1(x_2) + x_1 \\ y_2 = f_2(y_1) + x_2, \end{cases} \tag{4}$$

and finally, concatenate $y_1$ and $y_2$. The inverse operation can be computed with :

$$\begin{cases} x_2 = y_2 - f_2(y_1) \\ x_1 = y_1 - f_1(x_2). \end{cases} \tag{5}$$

These invertible equations illustrate how Invertible Networks operate a bijection between their input and their output.

### 3.4   CONTINUAL LEARNING SETTING

We propose to specialize each Invertible Network to a specific class by training them to output a vector with small norm when presented with data samples from their class. Given a dataset $\mathcal{X}_i$ of class $i$ and an Invertible Network $f_i$, our objective is to minimize the loss $\mathcal{L}$ :

$$\mathcal{L}(\mathcal{X}_i) = \frac{1}{|\mathcal{X}_i|} \sum_{x \in \mathcal{X}_i} \|f_i(x)\|_2^2 \tag{6}$$

Once the training has converged, the weights of this network won't be updated when new classes will be added. At inference time, after learning $t$ classes, the predicted class $y^*$ for a sample $x$ is obtained by running each network and identifying the one with the smallest output :

$$y^* = \underset{y=1,...t}{\arg\min} \|f_y(x)\|_2^2 \tag{7}$$

As it is common practice in image processing, one can also use a preprocessing step by applying a fixed pretrained features extractor beforehand.

## 4 EXPERIMENTAL RESULTS

We compare our method against several state-of-the-art baselines for single-head learning on MNIST and CIFAR-100 datasets.

### 4.1 IMPLEMENTATION DETAILS

**Topology of OvA-INN** Due to the bijective nature of Invertible Networks, their output size is the same as their input size, hence the only way to change their size is by changing the depth or by compressing the parameters of the intermediate networks $f_1$ and $f_2$. In our experiments, these networks are fully connected layers. To reduce memory footprint, we replace the square matrix of parameters $W$ of size $n \times n$ by a product of matrices $AB$ of sizes $n \times m$ and $m \times n$ (with a compressing factor for the first and second block $m = 16$ for MNIST and $m = 32$ for CIFAR-100). More details on the memory cost can be found in Appendix A.

**Regularization** When performing learning one class at a time, the amount of training data can be highly reduced: only 500 training samples per class for CIFAR-100. To avoid overfitting the training set, we found that adding a weight decay regularization could increase the validation accuracy. More details on the hyperparameters choices can be found in Appendix B.

**Rescaling** As ResNet has been trained on images of size $224 \times 224$, we rescale CIFAR-100 images to match the size of images from Imagenet.

### 4.2 EVALUATION ON MNIST

We start by considering the MNIST dataset (LeCun et al., 1998), as it is a common benchmark that remains challenging in the case of single-head Continual Learning.

**Baselines**

Generative models:
- Parameter Generation and Model Adaptation (PGMA) (Hu et al., 2018)
- Deep Generative Replay (DGR) (Shin et al., 2017)

Coreset-based models:
- iCaRL (Rebuffi et al., 2017)
- SupportNet (Li et al., 2018)

For Parameter Generation and Model Adaptation (PGMA) (Hu et al., 2018) and Deep Generative Replay (DGR) (Shin et al., 2017), we report the results from the original papers; whereas we use the provided code of SupportNet to compute the results for iCaRL and SupportNet with the conventional architecture of two layers of convolutions with poolings and a fully connected last layer. We have also set the coreset size to $s = 800$ samples.

Table 1: Comparison of accuracy and memory cost in number of parameters (and memory usage for storing samples if relevant) of different approaches on MNIST at the end of the Continual Learning. The Learning type column indicates the number of classes used at each training step.

| Model | Accuracy (%) | Memory cost | Learning type |
|---|---|---|---|
| PGMA (Hu et al., 2018) | 81.7 | 6,000k | 2 by 2 |
| SupportNet (Li et al., 2018) | 89.9 | 940k | 2 by 2 |
| DGR (Shin et al., 2017) | 95.8 | 12,700k | 2 by 2 |
| iCaRL (Rebuffi et al., 2017) | 96.0 | 940k | 2 by 2 |
| **OvA-INN (this work)** | **96.4** | **520k** | **1 by 1** |

**Analysis**

We report the average accuracy over all the classes after the networks have been trained on all batches (See Table 1). Our architecture does not use any pretrained features extractor common to every classes (contrarily to our CIFAR-100 experiment) : each sample is processed through an Invertible Network, composed of two stacked invertible blocks.

Our approach presents better results than all the other reference methods while having a smaller cost in memory (see Appendix A) and being trained by batches of only one class. Also, our architecture relies on simple fully-connected layers (as parts of invertible layers) whilst the other baselines implement convolutional layers.

## 4.3 EVALUATION ON CIFAR-100

We now consider a more complex image dataset with a greater number of classes. This allows us to make comparisons in the case of a long sequence of data batches and to illustrate the value of using a pretrained features extractor for Continual Learning.

**Baselines**

Distance-based model:
- Nearest prototype : our implementation of the method consisting in computing the mean vector (prototype) of the output of a pretrained ResNet32 for each class at train time. Inference is performed by finding the closest prototype to the ResNet output of a given sample.

Generative model:
- FearNet (Kemker & Kanan, 2017) : uses a pretrained ResNet48 features extractor. FearNet is trained with a warm-up phase. Namely, the network is first trained with the all the first 50 classes of CIFAR-100, and subsequently learns the next 50 classes one by one in a continual fashion.

Coreset-based models:
- iCaRL (Rebuffi et al., 2017) : retrains a ResNet32 architecture on new data with a distillation loss.
- End-to-end IL (Castro et al., 2018) : retrains a ResNet32 architecture on new data with a cross-entropy together with distillation loss.

**Analysis**

The data is provided by batch of classes. When the training on a batch ($\mathcal{D}_i$) is completed, the accuracy of the classifier is evaluated on the test data of classes from all previous batches ($\mathcal{D}_1, ..., \mathcal{D}_i$). We report the results from the literature with various size of batch when they are available.

OvA-INN uses the weights of a ResNet32 pretrained on ImageNet and never update them. FearNet also uses pretrained weights from a ResNet. iCaRL and End-to-End IL use this architecture but retrain it from scratch at the beginning and fine-tune it with each new batch.

The performance of the Nearest prototype baseline proves that there is high benefit in using pretrained features extractor on this kind of dataset. FearNet shows better performance by taking advantage of a warm-up phase with 50 classes. Still, we can see that OvA-INN is able to clearly outperform all the other approaches, reaching 72% accuracy after training on 100 classes. We can see that the performances of methods retraining ResNet from scratch (iCaRL and End-to-End IL)

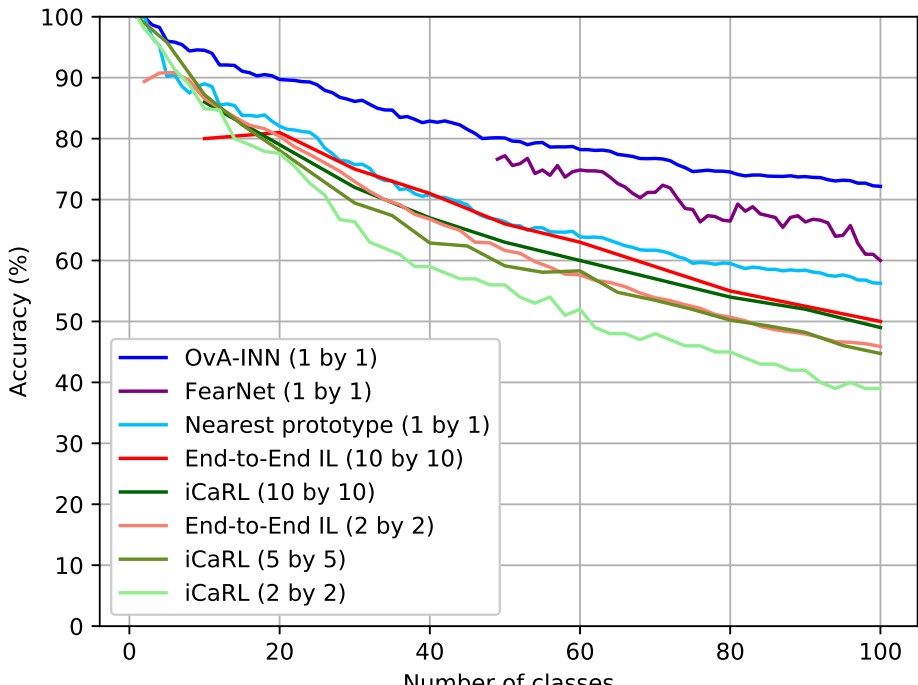

Figure 2: Comparison of the accuracy of several Continual Learning methods on CIFAR-100 with various batches of classes. FearNet's curve has no point before 50 classes because the first 50 classes are learned in a non-continous fashion.

quickly deteriorate compared to those using pretrained parameters. Even with larger batches of classes, the gap is still present.

It can be surpising that at the end of its warm-up phase, FearNet still has an accuracy bellow OvA-INN, even though it has been trained on all the data available at this point. It should be noted that FearNet is training an autoencoder and uses its encoding part as a features extractor (stacked on the ResNet) before classifying a sample. This can diminish the discriminative power of the network since it is also constrained to reproduce its input (only a single autoencoder is used for all classes).

To further understand the effect of an Invertible Network on the feature space of a sample, we propose to project the different features spaces in 2D using t-SNE (Maaten & Hinton, 2008). We project the features of the five first classes of CIFAR-100 test set (see Figure 3). Classes that are already well represented in a cluster with ResNet features (like violet class) are clearly separated from the clusters of Invertible Networks. Classes represented with ambiguity with ResNet features (like light green and red) are better clustered in the Invertible Network space.

## 5 DISCUSSION

A limiting factor in our approach is the necessity to add a new network each time one wants to learn a new class. This makes the memory and computational cost of OvA-INN linear with the number of classes. Recent works in networks merging could alleviate the memory issue by sharing weights (Chou et al., 2018) or relying on weights superposition (Cheung et al., 2019). This being said, we showed that Ova-INN was able to achieve superior accuracy on CIFAR-100 class-by-class training than approaches reported in the literature, while using less parameters.

Another constraint of using Invertible Networks is to keep the size of the output equal to the size of the input. When one wants to apply a features extractor with a high number of output channels, it can have a very negative impact on the memory consumption of the invertible layers. Feature Selection or Feature Aggregation techniques may help to alleviate this issue (Tang et al., 2014).

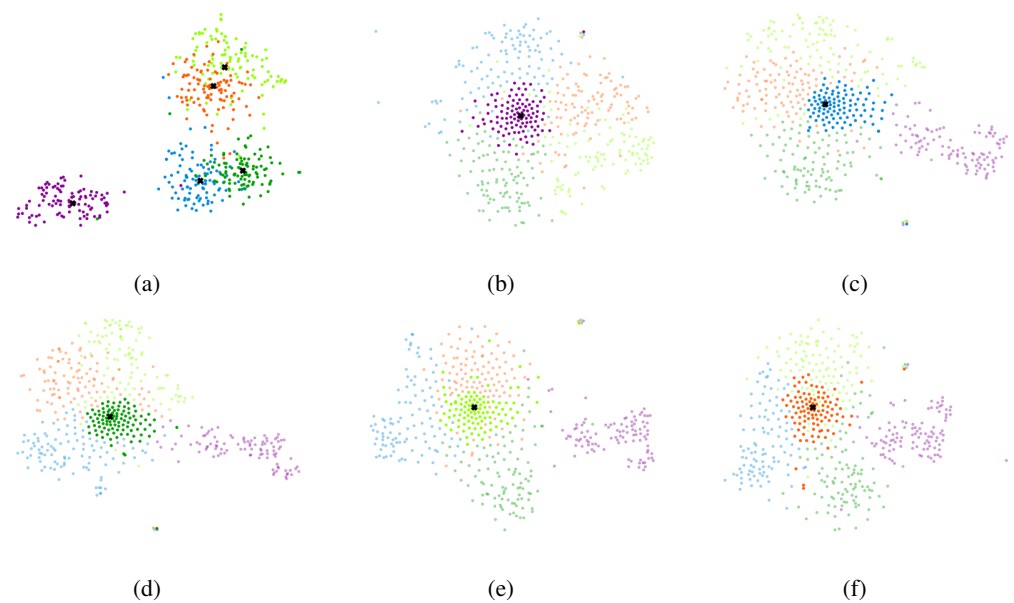

|     |     |     |
|:---:|:---:|:---:|
| (a) | (b) | (c) |
| (d) | (e) | (f) |

Figure 3: t-SNE projections of features spaces for five classes from CIFAR-100 test set (colors are given by the ground truth). *(a)*: features space before applying Invertible Networks (black crosses are the clusters centers). *(b),(c),(d),(e),(f)*: each features space after the Invertible Network of each class. The samples of a class represented by a network are clustered around the zero vector (black cross) whilst the samples from other classes appear further away from the cluster. Another visualization highlighting the differences between OvA-INN and Nearest Prototype is presented in Appendix D.

Finally, we can notice that our approach is highly dependent on the quality of the pretrained features extractor. In our CIFAR-100, we had to rescale the input to make it compatible with ResNet. Nonetheless, recent research works show promising results in training features extractors in very efficient ways (Asano et al., 2019). Because it does not require to retrain its features extractor, we can foresee better performance in class-by-class learning with OvA-INN as new and more efficient features extractors are discovered.

As a future research direction, one could try to incorporate our method in a Reinforcement Learning scenario where various situations can be learned separately in a first phase (each situation with its own Invertible Network). Then during a second phase where any situation can appear without the agent explicitly told in which situation it is in, the agent could rely on previously trained Invertible Networks to improve its policy. This setting is closely related to *Options* in Reinforcement Learning. Also, in a regression setting, one can add a fully connected layer after an intermediate layer of an Invertible Network and use it to predict the output for the trained class. At test time, one only need to read the output from the regression layer of the Invertible Network that had the highest confidence.

## 6 CONCLUSION

In this paper, we proposed a new approach for the challenging problem of single-head Continual Learning without storing any of the previous data. On top of a fixed pretrained neural network, we trained for each class an Invertible Network to refine the extracted features and maximize the log-likelihood on samples from its class. This way, we show that we can predict the class of a sample by running each Invertible Network and identifying the one with the highest log-likelihood. This setting allows us to take full benefit of pretrained models, which results in very good performances on the class-by-class training of CIFAR-100 compared to prior works.

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

## A  MEMORY USAGE

### A.1  MNIST

OvA-INN uses 2 blocks with 2 layers ($f_1$ and $f_2$) for 10 classes. The weight matrix of each layer $W$ is a product of two matrices $A$ and $B$ of size $392 \times 16$ and $16 \times 392$. The memory required for OvA-INN is :

$$\mathcal{S}_{\text{OvA-INN,MNIST}} = (392 \times 16 \times 2 + 392) \times 2 \times 2 \times 10 = 517440$$

We set the coreset size of iCaRL and SupportNet to $s = 800$ with each image of size $28 \times 28$, the convolutional network is composed of a layer of 64 channels with $5 \times 5$ kernel, a layer of 32

channels with $5 \times 5$ kernel, a fully-connected layer with 100 channels applied on an input of size $7 \times 7$ and a final layer of 10 channels :

$$\mathcal{S}_{\text{iCaRL,MNIST}} = 28 \times 28 \times 800 + (5 \times 5 + 1) \times 32 + (5 \times 5 + 1) \times 64 + (7 \times 7 \times 64 + 1) \times 100 + (100 + 1) \times 10 = 944406$$

### A.2 CIFAR-100

Since every method rely on a ResNet32 (around 20M parameters) to compute their features (except FearNet which uses ResNet48). We do not count the features extractor in the memory consumption.

OvA-INN uses 2 blocks with 2 layers ($f_1$ and $f_2$) for 100 classes. The weight matrix of each layer $W$ is a product of two matrices $A$ and $B$ of size $256 \times 32$ and $32 \times 256$. The memory required is :

$$\mathcal{S}_{\text{OvA-INN,CIFAR}} = (256 \times 32 \times 2 + 256) \times 2 \times 2 \times 100 = 6656000$$

We use the default coreset size $s = 2000$ of iCaRL and End-to-End IL with each image of size $32 \times 32$ :

$$\mathcal{S}_{\text{iCaRL,CIFAR}} = 32 \times 32 \times 3 \times 2000 = 6144000$$

## B  HYPERPARAMETERS SETTINGS

Our implementation is done with Pytorch (Paszke et al., 2017), using the Adam optimizer (Kingma & Ba, 2014) and a scheduler that reduces the learning rate by a factor of $0.5$ when the loss stops improving. We use the resize transformation from torchvision with the default bilinear interpolation.

Table 2: MNIST Hyperparameters

| Hyperparameter | Value |
| --- | --- |
| Learning Rate | 0.002 |
| Number of epochs | 200 |
| Weight decay | 0.0 |
| Patience | 20 |

Table 3: CIFAR-100 Hyperparameters

| Hyperparameter | Value |
| --- | --- |
| Learning Rate | 0.002 |
| Number of epochs | 1000 |
| Weight decay | 0.0002 |
| Patience | 30 |

Table 4: t-SNE Hyperparameters

| Hyperparameter | Value |
| --- | --- |
| Perplexity | 15.0 |
| Principal Components | 50 |
| Steps | 400 |

## C  TASK-BY-TASK LEARNING

We provide additional experimental results on the multi-head learning of CIFAR100 with 10 tasks of 10 classes each. The training procedure of OvA-INN does not change from the usual single-head learning but, at test time, the evaluation is processed by batches of 10 classes. The accuracy score is the average accuracy over all 10 tasks. We report the results from (Yoon et al., 2017). Although our approach is able to match state-of-the-art results in accuracy, it should be noticed that it is drastically more memory and time consuming than the other baselines.

| Model | Accuracy (%) |
| --- | --- |
| EWC (Kirkpatrick et al., 2017) | 81.34 |
| Progressive Networks (Rusu et al., 2016) | 88.19 |
| DEN (Yoon et al., 2017) | 92.25 |
| OvA-INN | 92.58 |

# D VISUALIZATION

We highlight the differences between OvA-INN and Nearest Prototype when classifying 20 classes of CIFAR-100 in Figure 4.

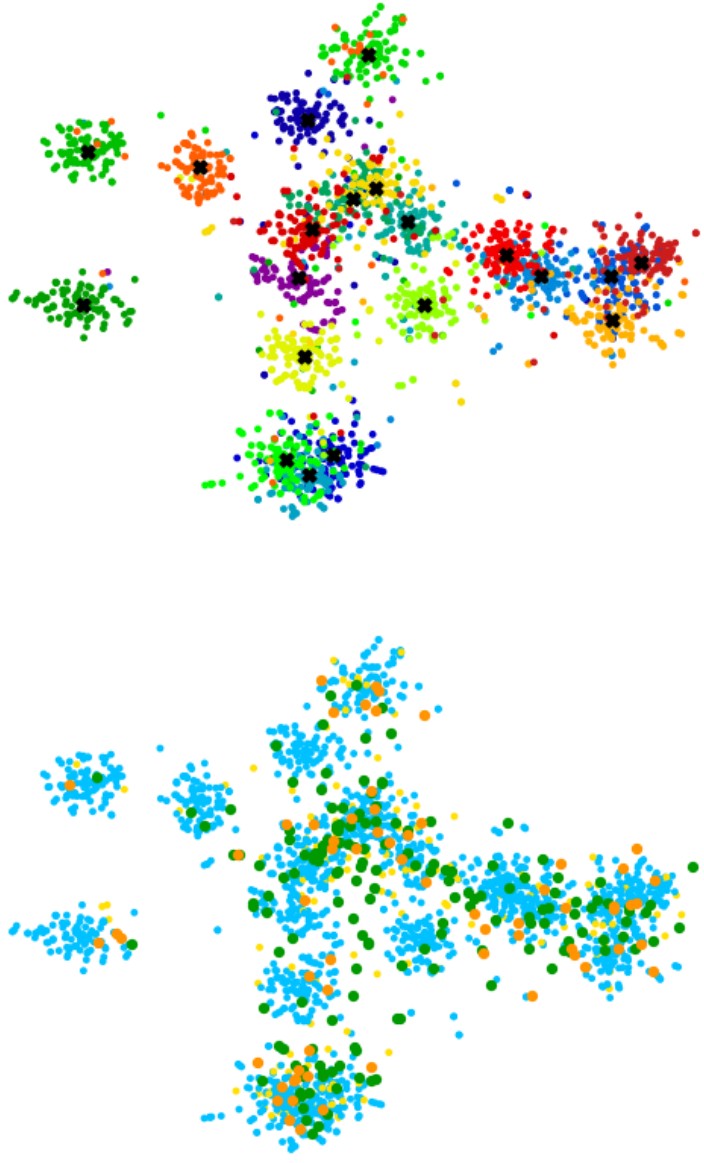

Figure 4: *top*: t-SNE projection of the features space before applying Invertible Networks (black crosses are the clusters centers) for 20 classes from CIFAR-100 test set (colors are given by the ground truth). *bottom*: in blue and yellow are the samples correctly and wrongly classified by both Nearest Prototype and OvA-INN, in green the samples better classified by OvA-INN than Nearest Prototype and orange the samples better classified by Nearest Prototype than OvA-INN.

