# OpenReview forum: "OvA-INN: Continual Learning with Invertible Neural Networks"
_ICLR.cc/2020/Conference — Reject_

### Official Review · AnonReviewer1 · 2019-10-22
**Official Blind Review #1**

**Rating:** 3

**Review:**

The central idea of this paper is to converts the classification problem into conditional generative modeling, it trains one normalizing flow (NICE) for each class of the data to perform out of distribution detection (one versus all binary classification). The data are presented one class at a time without being able to look back.

This paper proves a good improvement over the baseline methods compared to. Especially on CIFAR100, which is considered a difficult task in the regime of continual learning. It is quite evident to me that this approach has a big advantage over the other incremental classification methods.

I give a weak reject to this paper mainly because,

1. For invertible network, the NICE [1] coupling structure is used in this paper, although it is described in this paper as minimizing the distance from zero vector, the loss is basically equivalent to NICE with a Gaussian distribution as the target distribution. This is because normalizing flow maximizes the log p(z) + log(|det(z/x)|), in the volume preserving case the log determinant is a constant, it reduces to maximizing log p(z), which is equal to L2 loss if Gaussian distribution is chosen for p(z).

In my opinion, the paper could be better presented from the angle of generative modeling / out of distribution (OOD) detection. From this point of view, it would be desirable to include comparisons to the other OOD detection literature. It has been reported that the log probability is not a very good measure for OOD detection [2] because if an input has "simpler" patterns it would be classified as in distribution although it is actually OOD, and it has been reported that volume preserving flow (as used in this paper), also suffer from the same problem. There're several works proposing alternatives [3,4]. The good results of this paper may be because of the pretrained feature extractor, using a pretrained feature extractor could be practically useful but has limited insight for future research. Also, using a fixed feature extractor is avoiding the forgetting problem instead of solving it, it wouldn't generalize to more general / realistic continual learning setting.

2. The incremental classification setting in this paper is not a very practical assumption for continual learning. i.e. the task boundary are assumed to be available during training, and within each task the data are i.i.d shuffled. One of the goal of continual learning is to prevent catastrophic interference, training separate network for each task, in my opinion, is avoiding the problem rather than solving it. Although there're many papers in the continual learning regime assumes availability of task boundary, I think this setting doesn't bring too much insight to how we can eventually solve the general case continual learning. This been said, I think this paper could be better presented from the OOD detection point of view.


[1] NICE: Non-linear Independent Components Estimation https://arxiv.org/abs/1410.8516
[2] Do Deep Generative Models Know What They Don't Know? https://arxiv.org/abs/1810.09136
[3] WAIC, but Why? Generative Ensembles for Robust Anomaly Detection https://arxiv.org/abs/1810.01392
[4] DETECTING OUT-OF-DISTRIBUTION INPUTS TO DEEP GENERATIVE MODELS USING TYPICALITY https://arxiv.org/abs/1906.02994

**Experience Assessment:**

I have published one or two papers in this area.

**Review Assessment: Checking Correctness Of Derivations And Theory:**

I carefully checked the derivations and theory.

**Review Assessment: Checking Correctness Of Experiments:**

I assessed the sensibility of the experiments.

**Review Assessment: Thoroughness In Paper Reading:**

I read the paper at least twice and used my best judgement in assessing the paper.

---

> ### Author Response · Authors · 2019-11-14
> **Response to AnonReviewer1**
>
> Thank you for your very relevant remarks.
>
> 1. OOD detection point of view :
> We agree with you on the link with OOD detection and modified the Second 3 to present our approach through the lens of OOD detection with the NICE formulation.
>
> 2. Incremental classification setting :
> We do not think the choice of using of a pretrained features extractor is a simple way to avoid the problem of catastrophic forgetting. Even with relevant features, it is not straightforward to find a way to use them with no negative data (one can not retrain just like in transfer learning). That is why we proposed the Nearest prototype baseline, which was significantly exceeded by 16%; and the FearNet baseline (also using pretrained features), which was exceeded by 12%.
> Although the incremental classification setting is far from continual learning without task boundary, we believe that this is a relevant first step toward more real-world suited learning methods.

---

> > ### Comment · AnonReviewer1 · 2019-11-14
> > **About pretrained network**
> >
> > I think my point on the pretrained network is misinterpreted.
> >
> > I didn't mean that pretrained network is preventing forgetting, instead, I mean that the reason why OOD detection works well in this case is because it is trained on top of pretrained feature. Note that probability density based OOD detection does not work well if trained end to end as pointed out by [2].
> >
> > I agree that continual learning is a very difficult problem not straightforward solvable. But the task incremental setting (more precisely, incremental classification), in my opinion, is easy (The difficulty is on OOD detection itself) when casted to OOD detection problem. However, unlike other replay based method, the OOD detection approach can not generalized to other settings when the data does not come strictly one task at a time.

---

> > > ### Author Response · Authors · 2019-11-15
> > > **About the goal of the paper**
> > >
> > > We agree with you that training end-to-end OOD detection systems is not an efficient approach since (as you pointed out) it does not work well and is also very memory heavy. But the point of our paper is precisely to show that it is possible to perform Continual Learning without updating the whole network (which is in constrast with a significant part of the CL literature). We are aiming for a system that requires as few updates as possible to handle new classes for incremental classification.

---

### Official Review · AnonReviewer2 · 2019-10-23
**Official Blind Review #2**

**Rating:** 6

**Review:**

This paper tackles continual learning problem with stacks of invertible network blocks. Similar as the ensemble idea, the proposed method learns an invertible network for each new object class. During test, for each class, each learnt network outputs a norm and the one with the smallest norm is the predicted class. The proposed model has shown performance improvements in classification accuracy on MNIST and CIFAR100 datasets in incremental class tasks.

Despite the linear increase of memory usage over number of tasks as authors pointed out in the conclusion, this is an elegant method for continual learning problems in incremental class tasks. The paper is very-well written and easy to follow. To my best knowledge, the formulations in the paper are correct and clear. It seems that there are sufficient details for reproduction. However, I have the following concerns which, I think, may lower the contribution of the paper, unless authors can help clarify.

1. Since authors propose an invertible neural networks-based method for continual learning, instead of only focusing on incremental class, authors should also evaluate the proposed method in other continual learning tasks in object classification, for example, incremental domain and task, as defined in this review paper (https://arxiv.org/pdf/1810.12488.pdf).

2. Though I would assume OVA-INN would outperform state-of-the-art regularization-based methods, it would be convincing to show quantitative results of these methods (e.g. inclusion of Elastic Weight Consolidation in Figure 2).

3. I do not understand the term "learning type" in Column 4, Table 1. Please give the definition of this term.

4. Figure 3 is an interesting visualization of the latent representation learnt by the model. What about prototypical networks? It would be great to provide side-by-side comparison with such visualizations for the clusters learnt by prototypical networks the few-shot continual learning settings.

5. In Table 1, how much is the memory cost for nearest prototype method?

6. How much training data has been used over multiple classes?

7. Side note: it is unclear to me how this method can generalize to other tasks, e.g. regression problems and reinforcement learning problems. Authors can briefly discuss how the proposed method can be applied in these scenarios in the future work section.

**Experience Assessment:**

I have published one or two papers in this area.

**Review Assessment: Checking Correctness Of Derivations And Theory:**

I assessed the sensibility of the derivations and theory.

**Review Assessment: Checking Correctness Of Experiments:**

I assessed the sensibility of the experiments.

**Review Assessment: Thoroughness In Paper Reading:**

I read the paper at least twice and used my best judgement in assessing the paper.

---

> ### Author Response · Authors · 2019-11-14
> **Response to AnonReviewer2**
>
> Thank you for your review and your suggestions.
>
> 1. Experiments on other Continual Learning tasks :
> Although our method is specifically designed to perform class incremental learning, it is possible, as you noticed, to assess its performances in a task incremental setting by simply changing the evaluation protocol. We conducted the experiments on CIFAR100 with 10 tasks of 10 classes recognition and obtained an average accuracy of 92.58% over all 10 tasks. This result is on par with state-of-the-art [1], which is able to reach 92.25% accuracy on average. That being said, our method is clearly more memory consuming than a typical task-by-task approach. That is why we have added this result in a new Appendix, since it is not our primary objective.
>
> 2. Comparison with regularization-based methods :
> As you rightfully assumed, regularization-based methods perform very poorly in the incremental class learning setting. LwF.MC falls down to less than 10% accuracy on CIAFR100 (reported from Figure 2 (a) in [2]). And EwC  achieves less than 20% on MNIST (reported from Figure 4 (a) in [3]). Since the results of these methods are very low in our setting and in order to keep Figure 2 readable, we chose not to incorporate them.
>
> 3. The term "learning type" :
> This corresponds to the number of classes seen during each phase of training. We have made it clearer in the new version.
>
> 4. Visualizations in Figure 3 :
> The goal of the visualizations in Figure 3 is not so much to compare the differences in latent space representation between various methods as to illustrate the effect of INN on the clusters of data they have been trained on. We did not have the time to reproduce the experiments with Prototypical Networks, but it should have been very close to the visualization of the feature space before applying INN.
>
> 5. Memory cost for nearest prototype method :
> Table 1 presents results on MNIST. In this case, nearest prototype would consists in saving the mean of the raw data for each class. That would be 10 * 28 * 28 = 7840 parameters. But the accuracy would be very low. In the case of CIFAR100, that would be 100 * 512 = 51 200 parameters. Taking the Resnet34 features extractor into account, CIFAR100 nearest prototype has a cost in memory of 20 millions of parameters, compared to the cost of OvA-INN of 26 millions.
>
> 6. Training data used over multiple classes :
> We use all the training data available for each class, which corresponds to 500 training data for each class of CIFAR100, and around 6000 for each class of MNIST.
>
> 7. Generalization to other tasks :
> One can imagine a Reinforcement Learning scenario where various situations can be learned separately in a first phase (one INN per situation). Then during a second phase where any situation can appear without the agent explicitly told in which situation it is in, the agent can rely on previously trained INN to improve its policy. This setting is closely related to Options in Reinforcement Learning [4].
> In a regression problem, one can add a fully connected layer after an intermediate layer of the INN (not after the last layer because it is supposed to be close to a null vector on data from the relevant class) and use it to predict the output for the trained class. At test time, one only need to read the output from the regression layer of the INN with the smallest norm.
>
> [1] : Lifelong Learning with Dynamically Expandable Networks https://arxiv.org/abs/1708.01547
> [2] : iCaRL: Incremental Classifier and Representation Learning https://arxiv.org/abs/1611.07725
> [3] : Continual Lifelong Learning with Neural Networks: A Review https://arxiv.org/abs/1802.07569
> [4] : The Option-Critic Architecture https://arxiv.org/abs/1609.05140

---

### Official Review · AnonReviewer3 · 2019-10-24
**Official Blind Review #3**

**Rating:** 6

**Review:**

[update after rebuttal]

I thank the authors for their detailed reply, answers to my questions, and updates to the paper. (I especially appreciate the substantial effort to address all points and improve the manuscript even given the strongly negative rating. I am not personally in favour of "extremifying" the rating system the way ICLR did this year; I think it discouraged many authors from working on their rebuttals, and caused many reviewers to have a sense of inertia to keep their bad scores. I'm glad the authors didn't fall into the first camp, and I'm doing my best not to fall into the second! I think you did a very good job of improving the paper).

I think the manuscript has improved substantially; especially the clarifications about (not) using negative samples and additional detail on INNs with citations.

I find some of the sentences still unclear, and strongly suggest having the text read over for grammar/clarity for a camera-ready version.

But overall I recommend acceptance of this work.

------------
Paper summary:
This paper does continual learning using an invertible neural network (INN) trained to recognize each class. At test time a new example is presented to all of the INNs and the INN whose prediction has the smallest norm is used to predict the class of the test example. They do experiments on MNIST and CIFAR-100

Paper contributions:
- Review of methods and evaluation settings for continual learning
- Review of invertible neural networks
- Experiments comparing the proposed method to several other continual learning methods
- Examination of memory cost
- Exploration of feature space of trained INNs

Review summary & decision:
The proposed idea of using invertible networks for continual learning sounds interesting, and I think that this could be a good paper. The experimental evaluation is not consistent or complete enough for me to to tell, however, and the core motivation for the idea is not clearly explained. There are also some less critical, but still important aspects of the paper (related work, clarity of explanations, repetitiveness) which lead me to decide this paper is not currently ready for publication.

Reasons for decision:
1. A lot of statements and decisions are made without being explained, or are unclear / innacurate
      - In CL, the objective is to learn several tasks one after the other" this is not the objective, it's the problem setting. The objective is classification accuracy (or some other metric) in this setting.
      - The choice of datasets seems a little odd; why not CIFAR-10 as well?
      - It's very misleading to say your method is able to learn one class at a time without storing data; unless I've misunderstood this is only after pretraining all of the invertible networks with labelled data.
       - Why 500 examples per class for CIFAR-100, and how many for MNIST? What criteria was used to arrive at this number?
      - in the abstract you state the definition of continual learning says you can't have access to the data from previous tasks, but then mention that previous works use samples from previous data. This seems contradictory
      - I'm not extremely familiar with Li & Hoeim (learning without forgetting), but I don't think it's very accurate to call it distillation regularization
      - In 4.1 I don't think it's accurate to call this compression; it's just having smaller size layers in between (unless I misunderstood).
      - I'm not familiar with model superposition; it should be more explained since your claims of having lower memory cost rely on this, as far as I can tell.
      - The experimental baselines seem inconsistent / not comparable, making it difficult to evaluate what is going on. E.g. some update pretrained features and some do not. Some (including yours) do offline pretraining on a subset of examples, with inconsistent amounts of data, and sometimes this is considered part of training (Fearnet) and sometimes not (yours).
       - I don't understand the central motivation for using INNs. The only sentence that seems to talk about it is "this way ... network won't be able to have an output similar to the outputs on data from its training set", but this is not clear, not enough explanation, and seems somewhat innaccurate. It's not that it's not able, it's just not likely (I think; please correct me if I'm wrong!), and I don't see why this isn't also true of e.g. MLPs (i.e. for MNIST, train 10 MLPs each to classify [class i] vs [all classes except i] - In order to justify the proposed approach, the baseline described above (using MLPs instead of invertible NNs) would be very useful
 2. The training regime for the INNs is not clear (are they shown negative (other class) examples? How many, how are they sampled? I didn't find any hyperparameters for the INN training in the appendix, even though it says they should be there.
 3. The single head setting seems very similar to (or maybe the same as?) open set learning; I think this should be mentioned and works in this area should be reviewed.
 4. Section 3 is repetitive and unclear. It could be greatly shortened to make space for more experiments.
 5. It would be nice to see computational cost as well as memory cost; this is important in many settings where continual learning would be deployed.

Feedback/suggestions/nits (not necessarily part of decision assessment):
1. Cite the definition of continual learning (e.g. with a reference to a textbook or review)
2. A lot of the writing is unclear, wordy, and/or grammatically incorrect
    - inconsistent verb tense
         -  e.g. "if one would need .... it is required" should be "if one would need .... it would require" I'd suggest rewording this sentence entirely, because it's misleading - it says "retrain on this new dataset (which sounds like train just on the new data), but I guess you mean retrain on all data including the new data. e.g. "Updating the model with new data requires retraining on the full dataset (old + new data). However, there are.... method may not be applicable."
         - "we are reaching" -> we reach    - frequent use of "indeed" when it doesn't make sense
    - Section 3 repeats the intro, 3.1 and 3.2 are sort of saying the same thing, and are also sort of repetitive of the related work. The related work should all be in the related work section, and this section should just be about what _you_ are doing. Invertible neural networks should be reviewed in something like a 'background' section, which possibly could be in appendix. Right now that section has even more related work in it
3. Include tsne hyperparameters in appendix
4. "appendix" not "annexe" (annexes are on buildings :) )
5. when you define continual learning, bold it (not italicize)
6. "Y is then Y_i" -> Y = Y_i"
7. Caption for Figure 3 is unclear; I don't understand what I'm supposed to take away from these images. I also don't see a black cross.

Questions:
1. What is the motivation for using INNs specifically? (rather than e.g. normal MLPs).
2. What is the training regime for the INNs?
3. What is model superposition, is it expensive in computation or something other than memory, and if you don't do that what is the memory cost of your method?

**Experience Assessment:**

I have published one or two papers in this area.

**Review Assessment: Checking Correctness Of Derivations And Theory:**

N/A

**Review Assessment: Checking Correctness Of Experiments:**

I carefully checked the experiments.

**Review Assessment: Thoroughness In Paper Reading:**

I read the paper thoroughly.

---

> ### Author Response · Authors · 2019-11-14
> **Response to AnonReviewer3**
>
> Thank you for your detailed review.
> We shall begin by clarifying that, at training time, in our single-head Continual Learning setting, there is no negative data available. The data of each class is only provided for one class at a time and it is not allowed to store all the data of a class in subsequent training steps (some concurrent methods keep a small subset of old data, but OvA-INN does not store any data). We have done major changes in Section 3 to make it clearer.
>
> Reasons
>
> 1. Clarity of paper :
> We have not performed experiments on CIFAR10 because iCaRL and FearNet do not provide results on this dataset. Also, with its higher number of classes, CIFAR100 appears more relevant to measure the effect of catastrophic forgetting.
> In the CIFAR100 experiment, we used samples of 500 images because it corresponds to all the data available for one class (and around 6000 images for a class for MNIST).
> Some approaches rely on subsamples of previous data to mitigate catastrophic forgetting. We propose a method that don't rely on subsamples.
> In [1], the authors propose to use Knowledge Distillation loss as a penalty when learning new tasks, hence it acts as a regularization of the global goal of learning several tasks.
> In matrix compression, we mean that we represent a matrix with a smaller number of parameters. It does not strictly correspond to using two smaller layers since there is no bias and no activation function between the two matrices.
> Our experiments don't rely on model superposition, we are comparable in memory consumption without this. This is only a research direction that would even further increase our memory efficiency.
> There are various ways of handling Continual Learning. We agree with you that it does not facilitate comparison but we made our best to find a common evaluation setting and to illustrate the differences between each approaches. Our pretraining step only consists in using a model pretrained on a public dataset (namely Imagenet).
> Because INN are bijective, two different inputs can't have the same output. We don't use negative data, making the use of a MLP irrelevant for our approach.
>
> 2. Training regime :
> As previously stated, we don't use negative data. The hyperparameters are in Appendix B.
>
> 3. Difference between single-head Continual Learning and Open Set Learning :
> Open-set classification is a problem of handling unknown classes that are not contained in the training dataset. This is not the case in our study. Our method is not design to handle unseen classes at test time, it is made to be able to learn to recognize new classes when data from these new classes are given (without forgetting how to recognize previous classes).
>
> 4. Section 3 clarification :
> We have updated Section 3 in accordance to the recommendation of Official Blind Review #1. This new version presents our approach in more detail.
>
> 5. Computational Cost :
> You are right about the relevance of computational cost. We have emphasised the linear cost with the number of classes in the new section Discussion, but we are not able to provide quantitative comparison with other methods since we do not always have access to the source code. This being said, the computations of each INN can be done in parallel, which could be very efficient on the right architecture.
>
> Feedback
>
> 1. Definition of Continual Learning :
> The link to definition is in the first paragraph of the introduction, with the reference : Parisi et al., 2018. [2]
>
> 2-6. Thank you for your feedback, we have updated the paper accordingly.
>
> 7. Figure 3 :
> These plots illustrate the ability of an INN to cluster the features from one class and leaving further away the features of the other classes even though it has not been trained on these other classes. The black crosses are located around the center of each cluster.
>
> Questions
>
> 1. What is the motivation for using INNs specifically? (rather than e.g. normal MLPs).
> INN allow us to train on new classes data without any negative samples from other classes.
>
> 2. What is the training regime for the INNs?
> For each INN : train on samples from only one class, then don't update it anymore.
>
> 3. What is model superposition, is it expensive in computation or something other than memory, and if you don't do that what is the memory cost of your method?
> We did not implement model superposition. It would require to modify the input of each layer at a cost of an element-wise matrix multiplication at each layer. It is an example of research direction that would benefit to OvA-INN in making it more memory efficient.
>
>
> [1] : Learning without Forgetting https://arxiv.org/abs/1606.09282
> [2] : Continual Lifelong Learning with Neural Networks: A Review https://arxiv.org/abs/1802.07569

---

### Decision · Program_Chairs · 2019-12-19

**Decision:**

Reject

**Comment:**

This paper is board-line but in the end below the standards for ICLR. Firstly this paper could use significant polishing. The text has significant grammar and style issues: incorrect words, phrases and tenses; incomplete sentences; entire sections of the paper containing only lists, etc. The paper is in need of significant editing.

This of course is not enough to merit rejection, but there are concerns about the contribution of the new method, experiment details, and the topic of study. The results are reported from either a single run or unknown number of runs of the learning system, which is not acceptable even if the we suspect the variance is low. The proposed approach relies on pre-training a feature extractor which in many ways side-steps the forgetting/interference problem rather than what we really need: new algorithms that processes the training data in ways the mitigate interference by learning representations. In general the reviewers found it very difficult to access the fairness of the comparisons dues do differences between how different methods make use of stored data and pre-training. The reviewers highlighted the similarity between the propose approach and recent work in angle of generative modeling / out of distribution (OOD) detection which suggests that the proposed approach has limited utility (as detailed by R1) and that OOD baselines were missing. Finally, the CL problem formulation explored here, where task identifiers are available during training and data is i.i.d, is of limited utility. Its hard to imagine how approaches that learn individual networks for each task could scale to more realistic problem formulations.

All reviewers agreed the paper's experiments were borderline and the paper has substantial issues. There are too many revisions to be done.